# Evolution of river regime in the Mekong River basin over eight decades and role of dams in recent hydrologic extremes

Huy Dang[1] and Yadu Pokhrel[1]

[1]Department of Civil and Environmental Engineering, Michigan State University, East Lansing, 48823, United States

*Correspondence to*: Yadu Pokhrel (ypokhrel@msu.edu)

**Abstract.** Flow regimes in major global river systems are undergoing rapid alterations due to unprecedented stress from climate change and human activities. The Mekong River Basin (MRB) was, until recently, among the last major global rivers relatively unaltered by humans, but this is changing alarmingly in the last decade due to booming dam construction. Numerous studies have examined the MRB's flood pulse and its alterations in recent years; however, a mechanistic quantification at the basin scale attributing these changes to either climatic or human drivers is lacking. Here, we present the first results of the basin-wide changes in natural hydrologic regimes in the MRB over the past eight decades and the impacts of dams in recent decades by examining 83 years (1940-2022) of river regime characteristics simulated by a river-floodplain hydrodynamic model that includes 126 major dams in the MRB. Results indicate that while the Mekong's river flow has shown substantial decadal trends and variabilities, the operation of dams in recent years is causing a fundamental shift in the seasonal volume and timing of river flow and extreme hydrological conditions. Even though the dam-induced impacts are small so far and most pronounced in areas directly downstream of major dams, dams are intensifying the natural variations in the Mekong's mainstream wet season flow. Further, the additional 65 dams commissioned since 2010 have exacerbated drought conditions by substantially delaying the MRB's wet season onset, especially in recent years (e.g., 2019 and 2020) when the natural wet season durations are already shorter than in normal years. Further, dams have shifted up to 20% of the mainstream annual volume between dry and wet seasons in recent years; while this has minimal impact on the MRB's annual flow volume, the flood occurrence in many major areas of the Tonle Sap Lake and Mekong Delta have been largely altered. This study provides critical insights on the long-term hydrologic variabilities and impacts of dams on the Mekong's flow regimes, which can help improve water resources management in light of intensifying hydrologic extremes.

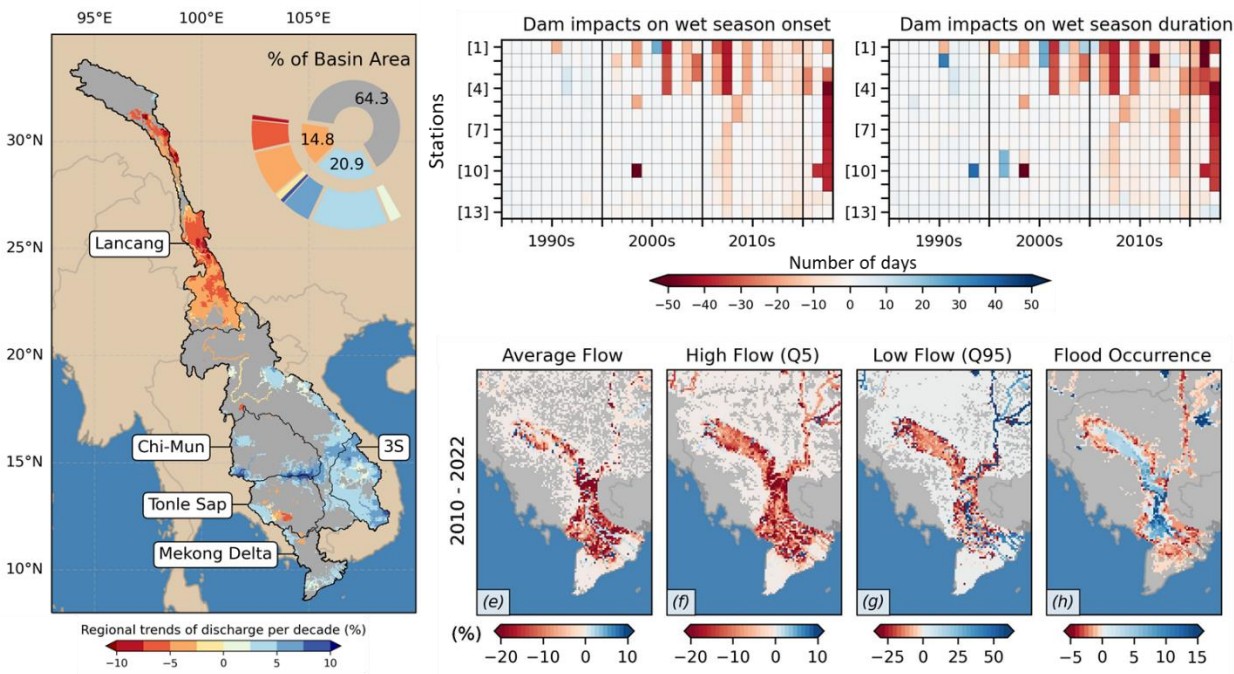

## 1 Introduction

A consistent pattern of river regimes is crucial in sustaining healthy hydrological and ecological systems in river basins (Botter et al., 2013; Bunn & Arthington, 2002; Poff et al., 2015). However, climatic and human drivers have been dramatically altering flow regimes in many global regions (Gudmundsson et al., 2021; Haddeland et al., 2014). For example, hydrologic extremes such as severe floods and droughts, which are being intensified by climate change (Calvin et al., 2023; Hirabayashi et al., 2010; Oki & Kanae, 2006; Pokhrel et al., 2021), are profoundly altering the hydrologic and hydrodynamic rhythms globally (Best, 2018; Grill et al., 2019; Nilsson et al., 2005). Such intensified climate extremes are causing more catastrophic floods and droughts, especially in densely populated regions with high flood-drought risk such as Southeast Asia (Lauri et al., 2012; Smajgl et al., 2015; Try et al., 2020; Västilä et al., 2010a; S. Wang et al., 2021). Humans have been using water infrastructures (e.g., dams) to reduce risks from such hydrologic extremes and better manage water resources. The construction of tens of thousands of dams globally (Lehner et al., 2011; Mulligan et al., 2020; Zhang & Gu, 2023) has greatly benefited our societies in reducing flood risk (Boulange et al., 2021); however, dams have been highly controversial (Flaminio et al., 2021; Graf, 1999) because large dams and their reservoirs fundamentally alter natural river regimes by redistributing water seasonally, causing detrimental ecological impacts (Best, 2018; Dethier et al., 2022; Ziv et al., 2012a). Yet, despite a slowdown in dam construction or even removal of existing dams in regions such as North America (Bednarek, 2001; Bellmore et al., 2017), dam building is booming in other regions such as the Mekong, Amazon, and Congo River basins (Winemiller et al., 2016; Zarfl et al., 2015).

In the Mekong River Basin (MRB), the alteration of river regime was historically small, at least in terms of mainstream Mekong flow (P. Adamson & Bird, 2010; P. T. Adamson et al., 2009; Grumbine & Xu, 2011); however, the acceleration in dam construction in the recent past and associated management of land and water systems (Cho & Qi, 2021, 2023) have led to rapid increase in the alteration of river flow and flood dynamics (M. E. Arias, Piman, et al., 2014; Chua et al., 2021; H. Dang et al., 2022; T. D. Dang et al., 2016). Being driven primarily by the Asian Monsoon, the MRB's hydrological rhythm is characterized by high, and rather unpredictable, seasonal variability (P. T. Adamson et al., 2009; Delgado et al., 2010a; J. Wang et al., 2022). Yet, the pattern of MRB's river flow seasonal cycle is remarkably consistent with a single, concentrated annual wet season which, on average, features throughout its 795,000 km$^2$ basin between approximately late June to early November (P. T. Adamson et al., 2009; Kummu & Sarkkula, 2008; Chua et al., 2021; Västilä et al., 2010). This leads to a prolonged flooding period in many parts areas of the Lower MRB, which is also known as the "flood pulse" (Pokhrel & Tiwari, 2022). During the remainder of the year, river flow gradually reduces to less than 10% (sometimes 5%) of its flood peak (P. T. Adamson et al., 2009). Additionally, the MRB has a distinct flow-reversal mechanism in the Tonle Sap River, whereby water flows into the Tonle Sap Lake (TSL) in Cambodia from the Mekong mainstream during the wet season, dramatically increasing the lake's size (by ~80%) (H. Dang et al., 2022; Kummu et al., 2013; Teh et al., 2019); the lake drains in the dry season, leading to a reversed flow in the Tonle Sap River (TSR) and supplying water to the Mekong Delta

(MD). Through this mechanism, the lake acts as a natural detention reservoir, creating a unique flood characteristic where areas between the TSL and MD are partially inundated for many months each year.

Owing to the unique and cyclic rhythm of the Mekong flood pulse, the river-floodplain ecosystems and local communities of the Lower MRB have been in harmony with the annual timing of this flood pulse. This flow rhythm supports fish migration and breeding including the seasonally flooded areas of the TSL (M. E. Arias et al., 2013; Baran & Myschowoda, 2009; Orr et al., 2012; Yoshida et al., 2020; Ziv et al., 2012b). Simultaneously, the flood pulse also brings rich nutrients each year in the form of sediment and a large volume of water to the floodplains in the areas between TSL and MD, which is critical to rice production. As a result, Cambodia has been ranked among the top countries for inland fishery production (Chea et al., 2023) while Vietnam is among the top rice exporters globally (Yuan et al., 2022). Additionally, the enormous water volume in combination with the mountainous topography in upstream areas is highly favorable for hydropower production, leading to the planning and construction of hundreds of dams in China, Laos, and Vietnam, especially in recent years (H. Dang et al., 2022; Shin et al., 2020). While an increased number of dams could be beneficial for flood control, the majority of Mekong's large dams are intended for hydropower production, which prioritizes power generation over downstream flood mitigation. Furthermore, these dams are physical barriers that directly hinder local fish migration and production annually (Chowdhury et al., 2024). As such, changes in the Mekong's river regime—especially the flood pulse—caused by intensified climate extremes and accelerating human activities could lead to potentially catastrophic impacts on the region's water, food, and energy systems and critical ecosystems.

With the critical role of the Mekong, the study of its hydrological attributes has been the focal point of both regional and global research for decades. Many studies have focused on the overall long-term trends of river flow (Delgado et al., 2010a; D. Li et al., 2017) and the patterns of the Mekong flood pulse, especially its timing and water budget (P. T. Adamson et al., 2009). Others have assessed the ecological impacts of changes in this flood pulse (M. E. Arias, Cochrane, Kummu, et al., 2014). Furthermore, intensified extreme floods and droughts (Keovilignavong et al., 2021) and the role of rapid hydropower development across the MRB in recent years (J. Gao et al., 2021; Ngor et al., 2018; Pokhrel & Tiwari, 2022) have captivated the attention of many investigations, leading to an increase in studies on the impact of these events. Overall, the changes in the Mekong flood characteristics have been the subject of numerous studies (Västilä et al., 2010b; J. Wang et al., 2021), especially on the impact of dams on river flow and inundation patterns (H. Dang et al., 2022; Pokhrel, Burbano, et al., 2018; Shin et al., 2020; W. Wang et al., 2017) as well as other human activities (M. E. Arias et al., 2012; Kummu & Sarkkula, 2008; NG & Park, 2021).

While past studies have provided important insights into the MRB's hydrological regime, there are notable limitations and major scientific gaps. First, many previous studies have relied on observed hydrological data which is limited to only a handful of stations in the Mekong mainstream (P. Adamson & Bird, 2010; P. T. Adamson et al., 2009; Delgado et al., 2010b), with considerable temporal gaps. Remote sensing-based datasets have helped overcome this issue to a certain extent, providing enhanced spatial coverage; however, they are available only for recent decades, often at a monthly scale, and remote sensing products generally suffer from uncertainties from various sources including cloud contamination (Bryant et

al., 2021; Lakshmi et al., 2023; Vu et al., 2021). As a result, there is a lack of analyses on the long-term trend of river flow across the entire MRB by using a spatially complete and temporally continuous dataset. Second, it is not possible to separate the impacts of dams from natural trends and variabilities in hydrologic extremes or the flood pulse by using only observed data for recent periods since there have been some large dams constructed in the MRB before the 1990s (H. Dang et al., 2022; Shin et al., 2020). Hydrological modeling can address this limitation; however, no studies to date have presented a full picture of the long-term hydrologic changes in the Mekong over the past century. Third, most studies on droughts in the MRB have focused primarily on the general drought indices and frequency (Y. Li et al., 2021; Lu & Chua, 2021; Tuong et al., 2021), while flood-related studies are more focused on changes only in the annual maximum flow (Chua et al., 2021; Delgado et al., 2010b; Västilä et al., 2010a), leaving critical research gaps regarding other aspects of these extreme events under the influence of both natural climate variability and dam operation.

In this study, we address the aforementioned gaps by applying a hydrodynamic model to simulate the hydrological attributes of the MRB over an 83-year period (1940-2022) and over the entire basin. We specifically address the following research questions. (1) How did the MRB's flow regime and flood pulse evolve over decadal time scales before and after the construction of major dams? (2) What are the relative impacts of dams compared to natural variabilities in the MRB's seasonal flows, hydrologic extremes, and inundation patterns in recent years? We address these questions by (a) examining the regional trend in river flow (i.e., annual total, maximum, and minimum) per decade across the MRB, and (b) attributing the observed trends and variabilities to natural variation and dam operation by comparing seasonal timings, flow volume and extreme conditions between simulations with and without dams. The remainder of the paper is organized as follows. Section 2 describes data and methods while results are presented in Section 3 accompanied by discussions in Section 4. Finally, Section 5 provides concluding remarks.

## 2 Data and Methodology

### 2.1. Data

Observed river flow and water level data used for model validation (see Sect. 3.1) at thirteen gauging stations on the Mekong mainstream (Fig. 1) are obtained from the Mekong River Commission (MRC). These stations are selected considering (i) a broad spatial coverage across the MRB and (ii) availability of at least 5 years of continuous observational data for both river flow and water level. Considering that there are temporal gaps in the observed dataset, model performance metrics were calculated only for periods for which observations are available for each station (Fig. 2). Additional information on the stations is provided in the supplementary information (Table S1).

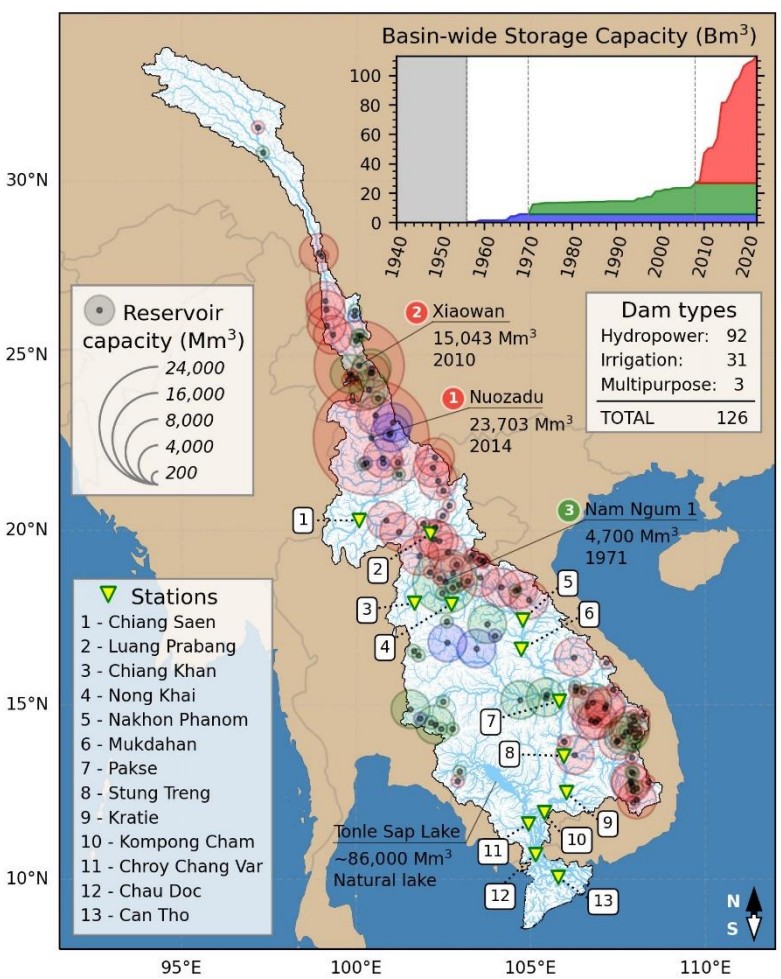

**Figure 1.** Spatial map of the MRB showing locations of the gauging stations (yellow triangles) used for validation and major dams (black dots) that are included in the simulation. Stations are numbered and their names are listed at the bottom left corner. Color coding and size of outer circles for each dam indicate the period of the dam commissioning and the reservoir's maximum storage capacity, respectively. The three largest reservoirs are numbered (color-coded circles) and named, along with their storage capacity and commissioned year; note that the color coding of the numbering for dams also indicates the period that the dam was commissioned. Tonle Sap Lake is also indicated. The background shows river network (blue lines) with thickness based on simulated long-term mean river flow from 1940 to 2022 at the 3-arcmin (~5km) spatial resolution. The basin-wide total reservoir storage capacity (billion cubic meters) for each year, color coded following dam's outer circles, is shown at the upper right inset with black dotted lines indicating the years that separate the periods.

Dam and reservoir specifications including coordinates, status (e.g., operational, planned, cancelled), year of commission, purpose (e.g., hydropower, multipurpose, irrigation, water supply), heights, storage capacity, or installed capacity are obtained from two primary sources: Research Program on Water, Land, and Ecosystem (WLE; https://wle-mekong.cgiar.org) which includes 445 dams and the Stimson Center (https://www.stimson.org/2020/mekong-infrastructure-tracker-tool/) which includes 777 dams in the MRB region. A comparison of the two datasets revealed that there is considerable inconsistency between them, including different dam specification values or duplicated dams under different

variations of their names. Additionally, a substantial number of dam specifications that are critical in simulation setups such as commission year, dam height, and reservoir storage were missing. Such inaccurate dam specification or the missing of a certain number of dams can yield inaccurate simulation results. Thus, we have synthesized the information on dam attributes for the entire MRB by building on the dam database prepared by Shin et al., 2020, and by combining additional information

from WLE and the Stimson Center. Specifically, we have filled any missing values and corrected erroneous records in the merged dam database using publicly available information collected from various resources including published reports from local governments or the MRC, documents from design and construction companies, other peer-reviewed literature as well as news articles. This results in a database of 693 dams in the Mekong region. Of these, 126 dams (compared to 86 in Shin et al. 2020 and Dang et al. 2021) commissioned by 2022 are selected based on criteria similar to our previous studies (H. Dang

et al., 2022; Shin et al., 2020): (1) dam height is at least 15 m (≥15 m), (2) storage capacity is over 1 million cubic meters (Mm$^3$), and (3) energy generation capacity is over 100 Mega Watts (MW). The location of these dams is shown in Fig. 1, and more information on dam specifications can be found in the supplementary information (Table S2).

## 2.2. Model and simulation settings

We use CaMa-Flood-Dam (CMFD), a river-floodplain hydrodynamic model that includes an optimized reservoir operation

scheme (H. Dang et al., 2022; Shin et al., 2020). This is an enhanced version of the Catchment-based Macro-scale Flood-plain model, CaMa-Flood (Yamazaki et al., 2011) version 4.0. The model discretizes the study domain into unit catchments, in which, each unit is assigned a set of river-floodplain topography parameters obtained from the MERIT Hydro dataset (Yamazaki et al., 2017) to present sub grid-scale hydrodynamic processes at ~5km (3-arcmin or 0.05°) resolution. Based on the unit's parameters and water storage, the model simulates river flow, water level, and inundated area following the local

inertial and mass conservation equations (Yamazaki et al., 2013). At unit catchments where reservoir outlets (or dams) are located, the natural outflow was recalculated based on the reservoir's designed purpose as follows: (1) at irrigation or water supply dams, dam release is simulated to meet downstream irrigation demand, and (2) at hydropower, the release amount is set to optimize power generation. Due to the lack of operating priorities for multipurpose dams in this region, these dams are represented in the model in a way similar to hydropower dams. More detailed information on the implementation of the

reservoir scheme can be found in our previous studies (H. Dang et al., 2022; Shin et al., 2020). Additionally, while water demand information is required for irrigation dam release calculation, there are no publicly available datasets for the MRB over the entire study period. Thus, we have applied the long-term seasonal average of the simulated irrigation demand from the HiGW-MAT model (Pokhrel et al., 2015), following our previous studies (H. Dang et al., 2022; Shin et al., 2020) as input in CMFD simulations.

CMFD simulations are driven by runoff data taken from the ECMWF Reanalysis version 5 (ERA5) global climate and weather dataset (Hersbach et al., 2020). We have selected the ERA5 dataset due to its (i) temporal completeness for our simulation period (i.e., 1940-2022) and (ii) higher spatial resolution (i.e., 0.25°) compared to other global forcing datasets used in our previous studies over the Mekong (e.g., Dang et al., 2022; Pokhrel, Shin, et al., 2018; Shin et al., 2020). This

approach of using global runoff datasets for CaMa-Flood simulations has been proven to yield good model performance in major global river basins (Chaudhari & Pokhrel, 2022; Shin et al., 2021; Tanoue et al., 2016; Yamazaki et al., 2012), especially the Mekong (H. Dang et al., 2022; Shin et al., 2020). However, initial results from ERA5 forcing indicated considerable overestimation of river flow at all stations upstream of Kompong Cham (Figs. S3-4). Thus, we applied bias correction to the ERA5 runoff dataset at basin scale as discussed in Sect. 2.3.

To quantify the effects of natural climate variation and reservoir operation on the MRB's hydrodynamic over the past decades, we conducted the following two simulations: (1) natural simulation without considering dams (NAT), and (2) regulated simulation by initiating dam operation from the start of their commissioned year (DAM). This results in 83 years of daily simulated river flow, water level, and flood depth for the entire MRB at the spatial resolution of ~5km (3-arcmin or 0.05°).

## 2.3. Data processing techniques and statistical measures

In climate impact studies, systematic deviations between simulated historical data and observations (precipitation, temperature, etc.) are commonly resolved by statistical and dynamical bias correction methods. However, to the authors' knowledge, studies with bias correction on runoff are scarce and uncommon because it is relatively difficult to collect runoff observations over large domains. Here, given substantial bias found in the simulated discharge when CMFD is forced with the ERA5 runoff data, we use runoff from the HiGW-MAT model—proven to yield accurate simulation results with CMFD for the MRB (H. Dang et al., 2022; Pokhrel, Shin, et al., 2018; Shin et al., 2020)—as a reference to bias correct the ERA5 runoff. We note that observed runoff data are not available for such bias correction at the basin scale and HiGW-MAT runoff could not be used because of its limited temporal coverage (1979-2016), especially for the purpose of examining extreme events in recent years. To preserve the general trend, variabilities, and extremes, while correcting the mean, standard deviation, quantiles, and frequencies of the ERA5 dataset, the Empirical Quantile Mapping (EQM) method (Mendez et al., 2020; Themeßl et al., 2012) was applied. First, the daily HiGW-MAT data were linearly interpolated from 0.5° to 0.25° to match the ERA5 resolution. Second, complete time series at each grid cell of HiGW-MAT data in the baseline period (1979-2016) was extracted to obtain its Cumulative Distribution Function (CDF). Similarly, two CDFs were obtained from ERA5 in each period (the baseline period and the remaining years). Third, values in ERA5 data during the baseline period were replaced by values with the same percentile in HiGW-MAT data. Fourth, we find the differences between HiGW-MAT and ERA5 values at each percentile in the baseline period. Then we applied these differences to the corresponding value in the ERA5 data based on their percentile from the CDF of the remaining years. The bias correction addresses the large overestimation found in the original ERA5-based results (Figs. S2-6), yielding notable improvements in the simulated river flow and water level across the MRB for periods both within (1979-2016; Figs. S3, 5) and outside (1940-1978; Figs. S4, 6) of HiGW-MAT data availability. Additionally, the results in Fig.S2 suggest that the combination of bias corrected ERA5 runoff and our dam scheme greatly improves the model's performance even at the daily scale. Thus, we use the results based on bias corrected ERA5 runoff for our analyses.

At each grid cell, the time series of simulated river flow is analysed to evaluate model performance, overall regional trends, and annual flood pulse characteristics. We first extracted time series data consisting of daily river flow and water level as well as total volume and maximum and minimum flow per calendar year. Daily and monthly simulated data is compared with observations using statistical measures such as volume changes (VOL), Nash-Sutcliffe coefficient (NSE) and Kling-Gupta efficiency (KGE). Then, the trend in flow at each cell was estimated using the Theil-Sen slope estimator (Gilbert, 1987), along with its statistical significance derived by applying the Mann-Kendall test (Mann, 1945). Additionally, various flood pulse characteristics, including the timing and magnitude of wet and dry season flow as well as annual extreme flows (maximum and minimum) using daily time series, were calculated. To detect the start and end of wet seasons, the long-term average river flow in the NAT simulation was applied as the season threshold following Adamson and Bird, 2010; Chua et al., 2021. Furthermore, we applied a seven-day moving average filter on the daily river flow in season timing analysis to avoid false detection of season onset due to early minor high-flow events (Fig. S1).

## 3 Results

### 3.1. Model performance

Results presented in Fig. 2—which also include statistical indicators—suggest that the model accurately reproduces the seasonal variations in river flow and water levels for most stations across the MRB. For river flow, the simulated results at all stations agree remarkably well with observations, especially given the size of the MRB and its hydrological and topographic complexities that are challenging to represent in basin-scale models. While there are small discrepancies between the simulated and observed river flow, the simulated annual volume (VOL) ranges between ~85-110% of the observed values, indicating slight overestimation (<10%) in stations upstream of Nakhon Phanom; in stations downstream of Pakse, the underestimations range from ~3% to 15%. Additionally, high NSE (0.74-0.92) and KGE (0.75-0.92) values at all stations with a wide range of observed data availability (AVL ranging from 7.4-99.7%) further confirm the accuracy of the model in capturing the natural variations in river flow. A similar observation can be made for the simulated water level at most stations where NSE and KGE values are relatively high (0.77-0.97 and 0.64-0.94, respectively) except for Pakse, Stung Treng, and Can Tho stations. The moderate performance at these stations could be attributed to uncertainties in the model's fixed parameters (e.g., channel width and depth) that are not specifically tuned as well as unaccounted human activities such as sand mining or other water infrastructure that could alter river morphology over time. The discrepancies in water levels could also be partly due to inconsistencies in the way water level is modeled and measured. For instance, the observed data are collected close to the riverbanks, which typically have a smaller difference between water surface to riverbed than the centre of the river, affecting water level readings. Considering that the river cross section is parameterized as rectangular (Yamazaki et al., 2011), simulated water level might include more discrepancies than river flow. Additional analysis on long-term trend of annual average, minimum and maximum river flow (Fig. S7-9) suggests that simulated results agree with observations at most stations with certain discrepancies especially when the detected trend is not statistically significant

(p>0.05). Since the primary objective of this study is to assess the annual and decadal variations in the hydrologic regime, these minor discrepancies are not of particular concern. Overall, a good model performance over a considerably long period (i.e., eight decades) supports model application to examine long-term evolution of hydrological conditions in the MRB and quantify dam impacts in recent periods.

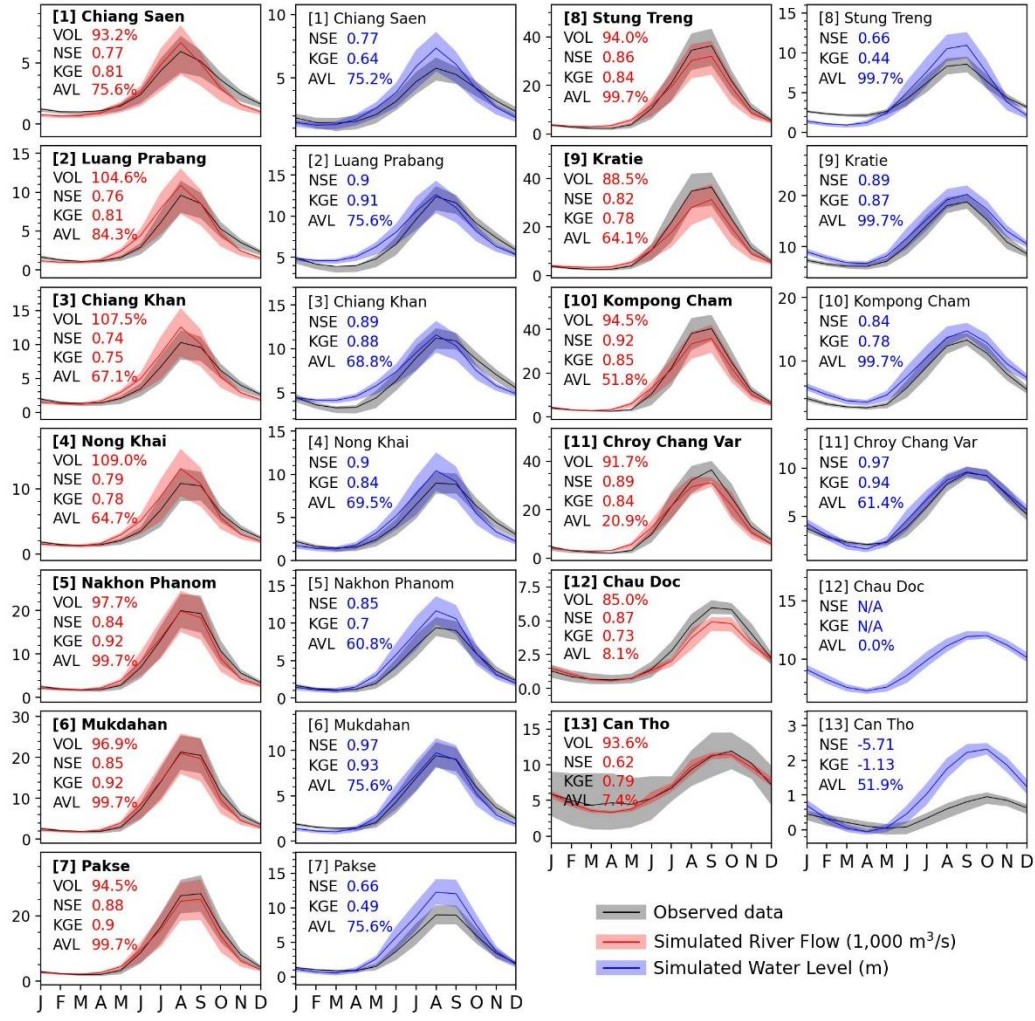

**Figure 2.** Comparison of seasonal cycle of simulated river flow (red lines) and water level (blue lines) with observations (black lines) from MRC at stations marked in Fig.1. Shadings of similar color coding indicate interannual variability presented as the upper and lower first quantiles for each month. Volume in percentage is indicated in panels with river flow validation while Nash-Sutcliffe coefficient (NSE), Kling-Gupta efficiency (KGE), and availability of observed data (AVL) in percentage are indicated in all panels.

### 3.2. Regional trends in river flow

Our results show a readily discernible regional pattern in river flow trend across the MRB (Fig. 3). The annual volume, maximum flow, and minimum flow show a varied spatial pattern with a general decrease in the Lancang portion and increase

in the lower portion of the MRB. In relation to mean annual volume, annual maxima, and annual minima of river flow over the 83-year period, regional trends typically vary within ±10% per decade (Fig. 3). Generally, the Mekong mainstream river flow is relatively stable with no distinct trend over the decades. In particular, only half of the total basin area shows a distinctly significant regional trend, mostly in the tributaries or subbasins. Local trends at the grid level range between -15% to +12% per decade for each of the flow characteristics. The spatial patterns of trends can be grouped into three main regions: a decreasing trend across all flow characteristics considered in the Upper Mekong (Lancang); an increasing but relatively mild trend in river flow in the mountainous areas of middle Mekong; and a mixed trend in the Sekong, Sesan, Srepok (3S) region in lower and Eastern parts of the basin.

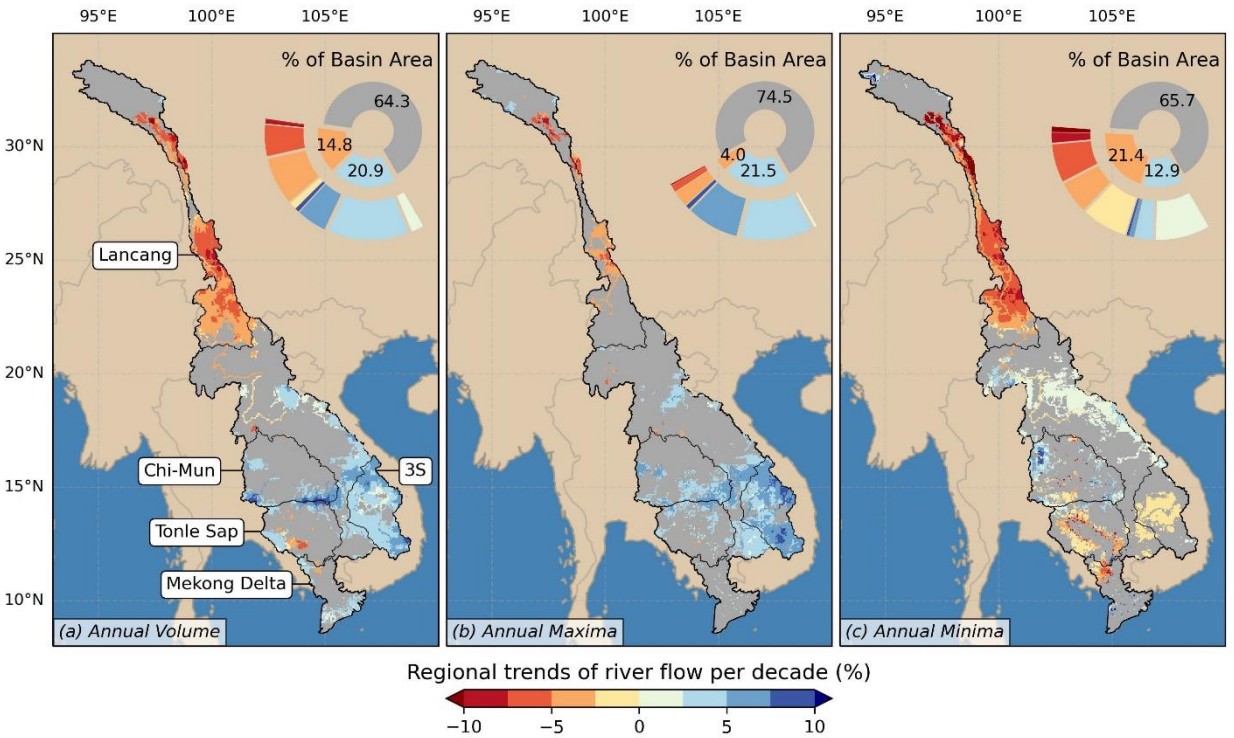

**Figure 3.** Decadal trends in simulated river flow (1940-2022), that are significant (p-value ≤ 0.05), shown for of (a) annual volume, (b) annual maxima (high flow), and (c) annual minima (low flow). Blue indicates increasing river flow, orange denotes decreasing river flow (in percent change of the long-term average values), and grey indicates areas with no significant trend. Similar color coding is applied to the inner pie chart in the top right corner, which indicates the percentage of basin areas which have respective trends. The outer semi-pie chart indicates a more detailed separation of areas with a significant trend following the color coding of the bottom color bar. Five major subbasins of the MRB including Lancang, Chi-Mun, 3S (Sesan, Srepok, and Sekong), Tonle Sap, and Mekong Mega-delta are named while their boundaries are indicated as thin black lines.

In terms of annual volume, ~36% of MRB area shows a significant trend, of which, 21% shows increasing trend while 15% shows decreasing trend. Most areas with decreasing trend are located in the Lancang region in the upstream, especially its middle portions where the decreasing trend is more pronounced (2.5-10%). In contrast, increasing trend in annual volume is mostly seen in the middle and lower MRB, specifically in the 3S subbasin and surrounding areas where a 2.5-7.5%

increase per decade is prominent. Additionally, the region at the border between Chi-Mun and TSL regions or Southwest of Chi-Mun shows a higher increase, with values that range from 5-10% in some locations. Mild increase can be seen in the middle of MRB and some coastal areas of the MD. Additional analyses comparing the decadal difference of our simulated annual volume and the ERA5 snowfall data (Figs. S10-11) suggest there is no clear linkage between a decline in annual volume to the changing snowfall pattern. However, there is a substantial resemblance in the pattern of decadal difference between ERA5 runoff, total precipitation (Figs. S12-13), and annual volume (Fig. S10), which further confirms that the annual volume in the Lancang area is also largely influenced by rainfall instead of snowfall.

In terms of annual extremes, only one-fourth of the MRB's total area shows significant spatial trend in annual maxima or flood peak, while this number in annual minima is approximately one-third. Out of the 25% area of Mekong with significant spatial trend in annual maxima, ~21% presents an increasing trend, located primarily in the lower Mekong and the 3S subbasin and its surroundings; these trend values range from 2.5-7.5% with some areas reaching to over 10%. While there are some signs of decreased flood peak in the Lancang region, these include small areas in the upper reaches where flooding is not prevalent. On the contrary, in the 34% of the Mekong area with significant trend in annual minima, there is ~21% of area that shows a decreasing pattern. Again, most of this decreasing trend is present in the Lancang region with a substantial drop from 2.5 to above 10% per decade. Surprisingly, the 3S region, which shows an increase in annual volume and annual maxima, also presents a minor drop of <2.5% per decade. Areas that are partially flooded in the outer areas of the TSL also witness a drop of annual minima flow between 2.5-5% per decade. Similar to annual volume, a slight increase in the middle of Mekong with 0-2.5% of annual minima per decade is also observed.

### 3.3. Natural variation and dam impacts on the flood pulse

### 3.3.1. Seasonal timing

Figure 4 presents a summary of the seasonal timing of various flow regimes (i.e., annual minimum, annual maximum, onset of wet season, and end of wet season) per calendar year, along with the variations in these attributes under natural drivers (i.e., climate variability) and dam operation over the past eight decades at selected stations. Figure 4a provides clear evidence that the overall hydrological timing of the Mekong is generally consistent across space (i.e., across stations in Fig. 4a) and time (i.e., temporal range for each attribute in Fig. 4a). All features of the seasonal timing across the stations from upstream to downstream only vary between two weeks to a month, which is in line with previous findings (P. T. Adamson et al., 2009). Typically, minimum flow occurs in March, while maximum flow occurs in between mid-August to mid-September. The wet season generally starts between mid-May to mid-June and ends in the first half of November. Additionally, there is a discernible delay in the timing of each attribute in the downstream stations, which ranges between a few days to a week compared to that for an immediately upstream station. However, the two most downstream stations in the Mekong Delta region (stations 12 and 13) show a distinct timing difference compared to other stations where all attributes are delayed by 2-4 weeks compared to only a few days for the upstream stations.

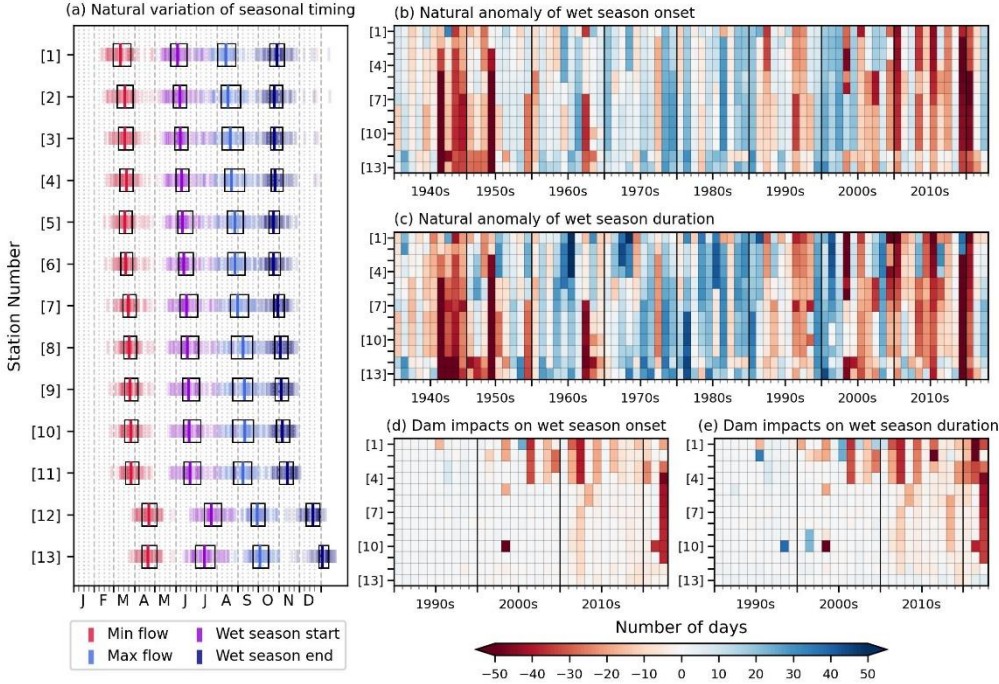

**Figure 4.** Variation in the seasonal timing of the flow regimes (a), anomaly of the onset (b) and duration (c) of the wet season driven by natural variations during 1940 to 2022, and impact of dams on wet season onset (d) and duration (e) during 1990-2022. Y-axes mark station numbers (as depicted in Fig. 1) for all panels. In panel (a) the timing of minimum flow, maximum flow, start of wet season, and end of wet season are indicated as thin color-coded lines, the black box indicates the 25th and 75th percentile, while the long-term median is shown as thick lines. Panels (b-e) share the same colorbar in the bottom right. Panels b and c show the difference in natural variation of wet season onset and duration, respectively, compared to the long-term average (in days). Panels (d,e) present the difference between DAM and NAT simulations for wet season onset and duration in recent years.

Comparison of the wet season onset and duration for each year to the long-term average (Figs. 4b-c) suggests that there is a high correlation between the two attributes. Similar anomalies for the two attributes (negative values indicating late onset or decrease in duration) suggest that the timing of the wet season onset can be a reasonable indicator of whether the wet season of that particular year will be reduced or extended compared to the long-term average. Furthermore, this alignment confirms that the ending of the wet season at upstream locations is relatively consistent, occurring at the beginning of November, regardless of the wet season onset being late or early. Additionally, the results further prove that there is a strong propagation effect from upstream to downstream Mekong despite the fluctuation in annual local precipitation patterns.

Results also show that the wet season onset has been significantly delayed with an alarmingly shorter length than the long-term average (by 20-50 days or higher) in recent years, especially in 2019 and 2020 (Figs. 4b-c). Lastly, close observation of the temporal evolution suggests that there is a notable interdecadal variation in the wet season timing, with noticeable period of late–short wet season in the 40s and 50s, followed by 3 decades of generally early–long wet season, and then followed by 3 decades of late–short wet season. We note that the results presented in Figs. 4b-c do not include any effects of dams, which are known to be prominent in the recent decade (discussed next). Overall, these results from the NAT

simulation evidently illustrate that the natural hydrologic regime of the MRB had substantial inter-annual and inter-decadal variations in terms of the onset and duration of the wet season, two crucial elements of stable hydrologic and ecological systems, especially in the downstream of the basin. The results also imply that there could be potentially enhanced variabilities in the future in the face of climate change and growing influence of dams.

We find that the construction of dams in the recent decades (since 1990s) has impacted the seasonal timings in a substantial way (Figs. 4d-e). Compared to the effect of natural variation, dams are generally delaying the wet season onset with only a few rare instances where the impacts are the opposite at some of the most upstream locations (e.g., 1996 and 2005; Fig. 4d). Similarly, wet season duration is also being reduced by dams with only some exceptions at specific stations and years (e.g., 1995, 1998, 2001, 2005; Fig 4e). Further, dam impacts are generally localized and more pronounced in the upstream areas (immediately downstream of the dams), with the delay ranging between 10-30 days in these locations. Due to the propagation effect of the river's seasonal cycle, upstream dam operation is expected to have a basin-wide impact. However, the delaying effect of upstream dams on downstream season timing is typically contained within a few stations and is detectable on a basin scale only in some years, especially after 2010, a period during which many mega dams were constructed. With booming dams in recent years, wet season is being delayed basin wide, and as a result, the wet season duration is being reduced. While the impact is only a few days of delay, in critical environmental and agricultural areas such as the TSL or the MD, these adverse effects are detected in years that have already seen a substantial delay due to natural variations. In brief, dams are exacerbating the high natural variability in the onset and duration of the wet season even though the impacts of dams are smaller so far and constrained within the river reaches with major dams.

More detailed analyses of the long-term natural river flow at locations along the Mekong mainstream and TSR (Fig. 5) suggest that the abrupt shift in seasonal timing at stations 12 and 13 compared to upstream locations is likely due to the natural retention reservoir effect of the TSL. This shift in seasonal timing is first detected in location directly downstream to station 11, the confluence of Mekong mainstream and TSR, and is carried on to all selected locations further downstream, deep into the MD. Since there are no available gauging stations between 11 and 12, 16, for further analyses, we have selected one location (marked as 15 - Unknown) directly downstream where the Mekong mainstream and Tonle Sap River meet. Hydrographs at various locations in Fig. 5b suggest that long-term river flow at stations 9 and 10 are almost identical while similar flow timing can be observed at station 11, with visibly lower peak, however, the river flow timing change immediately downstream to station 11, at location 15. Additionally, the hydrograph indicates that, in the first five months, TSR contributes substantially to downstream river flow especially in January and February, similar to findings in Morovati et al., 2023, where TSR outflow (station 14) is equal or even higher than the flow from Mekong mainstream (station 10). As a result, while Mekong upstream river flow falls to its minimum in late March, minimum flow at all locations downstream of station 11 occurs much later in the middle of April. Similarly, the delayed wet season end date can also be directly attributed to the influence of TSL as there is a clear difference of river flow before and after station 11 from late September, when water flows out of TSR.

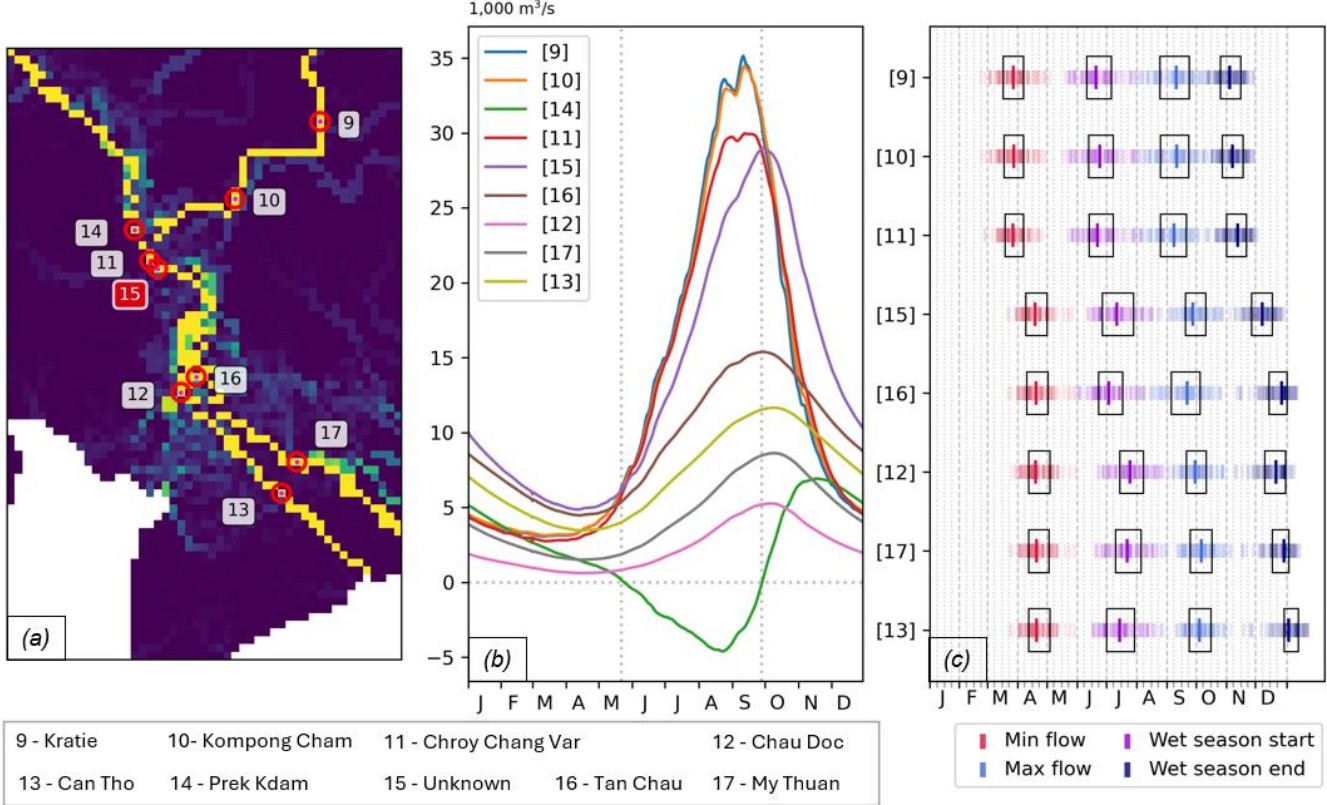

| 9 - Kratie | 10- Kompong Cham | 11 - Chroy Chang Var | 12 - Chau Doc |
|---|---|---|---|
| 13 - Can Tho | 14 - Prek Kdam | 15 - Unknown | 16 - Tan Chau | 17 - My Thuan |

**Figure 5.** Variation in the seasonal timing of the flow regimes at most downstream Mekong mainstream locations driven by natural variations from 1940 to 2022. In panel (a), the locations of selected stations are indicated as numbered red circles on top of long-term average simulated river flow. Panel (b) shows the color-coded long-term average simulated river flow at each selected location with vertical dotted lines indicating when the TSR changes its flow direction (outflow from TSL is positive, while inflow to TSL is negative). In panel (c), the variation in seasonal timing at all selected locations on the Mekong mainstream and channels (station 14 not included) are shown in similar color-coding and format to Fig. 4.

Lastly, the wet season of station 13 is further extended by a more than two weeks into January of next year compared to that for station 12. Additional analysis at similar locations of stations 12 and 13 on the other main channel of Mekong in MD (Fig. 5c) suggests that the end of wet season gradually shifts to later dates as we move from station 11 into MD on both Mekong main channels. A direct comparison in seasonal timing between location 15 with station 12 and 16 suggests that this prolonged effect is likely due to the river being divided into multiple channels instead of direct influence from TSL. When the river diverges into multiple channels, the progression of river flow in each channel becomes relatively stable, with less dramatic rises and drops from peak flow as can be seen in the hydrographs, resulting in a longer wet season. This is further confirmed by a clear difference between seasonal timing within a channel where there are more visible divisions (i.e., channel with stations 12 and 13) than the other (channel with stations 16 and 17) in the MD as shown in Fig. 5. Thus, while the abrupt shift in seasonal timing of the most downstream Mekong areas can be directly attributed to the TSL influence, the prolonged wet season effect in the MD can be attributed to the rather flat topography and the extensive irrigation channel

network in this area. However, it should be noted that the representation of the river channel network and other water infrastructure (i.e., dikes) in the current model for this region is partially incomplete due to multiple limitations, thus, we expect the actual wet season-prolonging effect of the MD channel network to be even more substantial.

**3.3.2. Water balance: natural interdecadal variability and dam impacts**

To examine the impact of natural climate variability on the water balance at the Mekong mainstream stations, annual flow
volumes for the last three decades at each station are compared to the long-term average (Fig. 6a). First, the model distinctly captures the anomalously dry (e.g., 1998, 2015, and 2019-2021) and wet (e.g., 2000 and 2011) years, discussed in previous studies (Pokhrel, Shin, et al., 2018; Shin et al., 2020). Second, while substantial interannual and interdecadal variability can be observed (Fig. 6a), it is clearly discernible that the MRB entered a prolonged water-deficit period starting in the mid-2000s, which intensified largely in recent years. Since 2004, results indicated multiple consecutive years with annual
volumes well below average (e.g., ~10%) across the basin. This period reached its peak in 2019 (a major drought) with ~20 to over 40% decrease in annual volume at different stations. This is followed by two consecutive drought years with more than 20% lower volume than average across the entire basin; a sign of recovery started showing in 2022, especially in downstream areas. Overall, these results indicate that there is a generally consistent tendency of decline in annual volume due to climate variation in the recent decades compared to long-term mean, which holds for all stations examined (Fig. 6a).

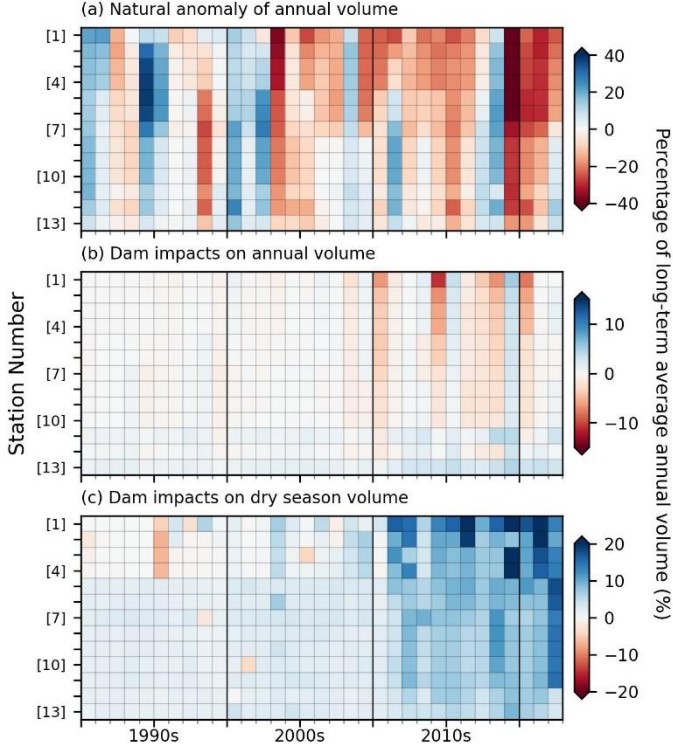

**Figure 6.** Anomaly of natural annual volume compared to the long-term (1940-2022) annual average (a), changes in each year's annual volume due to dam impacts (b), change in dry season volume due to dam operation (c). All results are normalized by the long-term annual volume at that station and then converted to percentage values. Y-axes mark station numbers (as depicted in Fig. 1) for all panels.

Figure 6b depicts how dams are affecting the annual volume over time and at different locations along the mainstream, indicating more pronounced impacts since 2010. Evidently, there are signs of dam impacts at an interannual scale, causing a ~5-10% decrease in the long-term average volume from one year to the next, especially in the upstream areas. However, a mild increase of <5% can be observed in recent years at the downstream areas (stations 9-13), which is due to dam operation in the Lower MRB tributaries (e.g., the 3S region) . Evidence from comparing the effects on annual volume between Fig. 6a and 6b confirms that the magnitude of dam impacts on annual volume are not substantially higher than the natural variabilities, especially in dry years. In terms of seasonal volume difference due to dam operation (Fig. 6c), results suggest that there has been relatively small impact (<5%) in previous decades (e.g. 1990s, 2000s). However, the shift in seasonal volume from the wet season to the dry season (i.e., the difference in dry season volume between DAM and NAT simulations) has increased dramatically from <5% to ~10-20% across the Mekong mainstream since 2011 with areas in the upstream witnessing an increase by over 30% of the long-term average volume. These results illustrate that the impacts of dams have become more prominent in recent years in terms of both annual and dry-season flow volumes, which may have profound implications on downstream hydrologic, agricultural, and ecological systems.

### 3.4. Dam impacts on extreme events and flooding pattern

Figure 7 presents the decadal average of the difference between DAM and NAT simulations for annual average flow, high flow, low flow and the flood occurrence in the TSL and MD areas. Additionally, a map of the mainstream grid cell can be found in the supplementary (Fig. S14). Result reveals that even in the 2000s, upstream dam operation had already caused visible impact to all flow regime attributes in the most downstream areas of the Mekong. In agreement with Fig. 6b, Fig. 7a suggests that the mainstream average flow in the 2000s is relatively unchanged compared to the long-term average. However, average flow over floodplain areas in the proximity of the Mekong mainstream and the TSL decreased marginally (<10%). Similarly, high flow in the mainstream shows a minor decrease (<2%) while this decrease is substantially higher in the surrounding floodplains (~5-10%; Fig. 7b). Additionally, further confirming dam-induced increase in dry season flow seen in Fig. 6c, Fig. 7c reveals that this effect was moderate in the mainstream during the 2000s. In contrast, low flow in the floodplain decreased similarly to the average and high flows. These results are closely related as flow in the floodplain areas is typically generated during high flow or flooding events, thus, a decrease in flood peak due to dam operation directly caused a decrease in river channel overflowing, effectively reducing flow in the floodplains and consequently, flood occurrence in these areas. Previous studies (H. Dang et al., 2022) suggested that upstream dam operation is shrinking the TSL by reducing flood occurrence in the lake's outer areas. However, our results in Fig. 7d suggest that this effect is not limited within the TSL but further propagates downstream to the MD where the outer areas of the mainstream also witness a decrease in flood occurrence (<5% or ~18 days per year). Additionally, due to the increased low flow, flood occurrence in the floodplain areas near the mainstream between TSL and MD increased (~5%) during the 2000s (Fig. 7d). In the 2010s, similar dam-induced impacts can be observed; however, the magnitude of these effects abruptly increased by two times or higher. The mainstream's average and high flow remained relatively unchanged with only a small difference (<3%); however, its low flow increased substantially (by ~50%) compared to the long-term average in some areas while all flow attributes in the majority of the floodplain decreased substantially (~20%). The effect of dam operation on the flood occurrence between TSL and MD also increased as flooding in the inner areas was further prolonged by 10-15% (~36-55 days), while the outer area flooding was diminished. That is, dam operation has largely altered the seasonality of river regime in this region, and subsequently changed the inundation patterns in the TSL and MD areas. This is concerning for river-floodplain ecosystems and local communities considering that the 2010s was already a historically dry decade for the MRB.

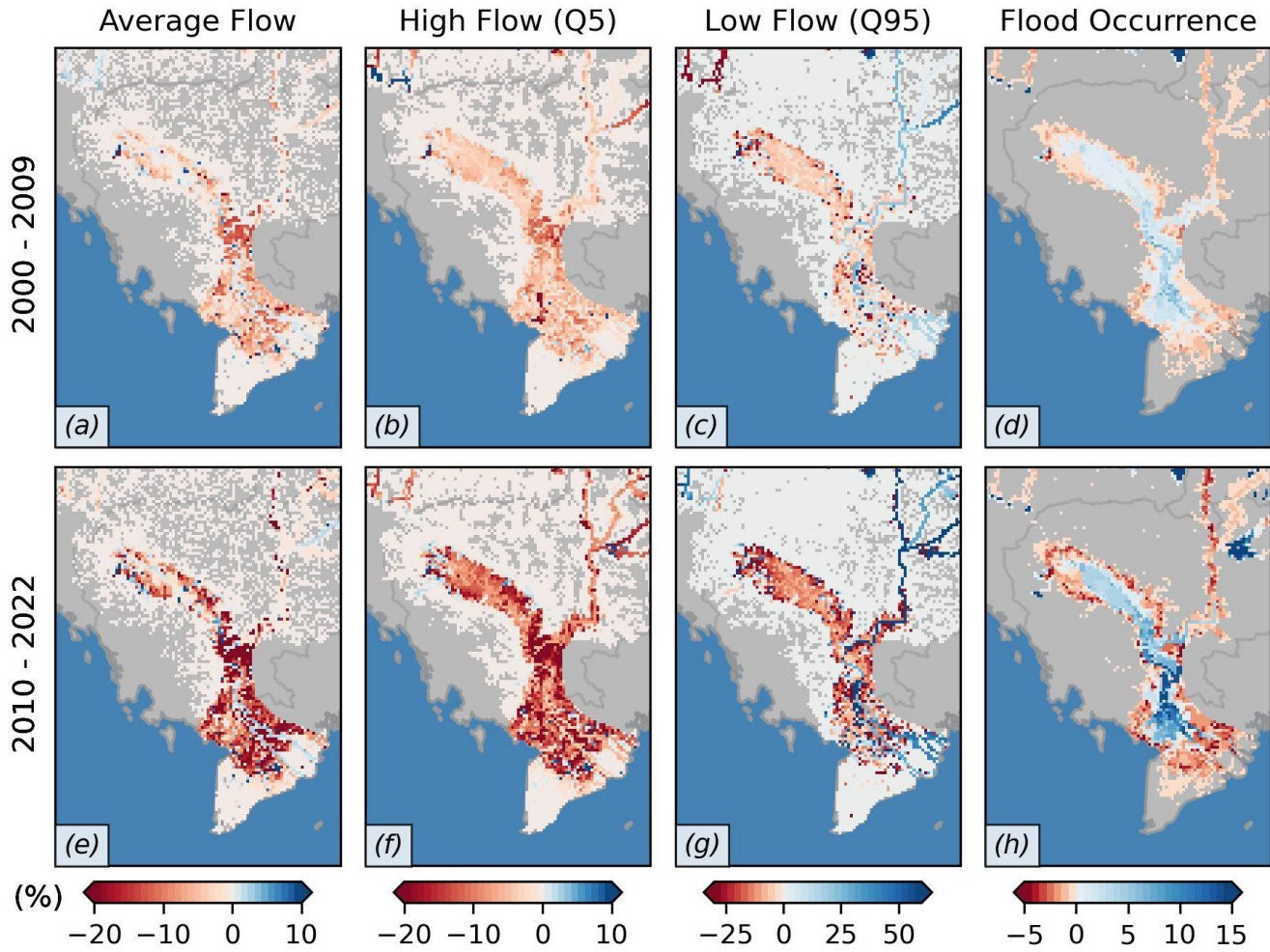

**Figure 7.** Differences in flow regime attributes between the DAM and NAT simulations (period indicated) relative to the long-term (1940-2022) average of the NAT simulation over the Tonle Sap and Mekong Dela areas including (a, e) average flow, (b, f) high flow (Q5), (c, g) low flow (Q95). Panels d and h show the difference in flood occurrence (percentage change) due to dam operation for the period indicated. Results for two periods are shown: 2000-2009 (a-d) and 2010-2022 (e-h). Areas outside of the MRB or having no changes are indicated in dark grey. Note that the colorbar range is difference among the panels.

## 4. Discussion

The changing hydrology of the Mekong River basin has been examined by numerous studies using various techniques and datasets; however, an in-depth analysis of the long-term (e.g., decadal and inter-decadal) trends in basin-wide river flow regime attributes has never been reported. Furthermore, the impacts of dams on the Mekong flood pulse and extreme events are generally studied separately and only assessed over short time periods. In this study, we use a combination of state-of-the-art modeling approaches and data analysis techniques to mechanistically quantify the changes in various river flow characteristics and flood occurrences, and attribute those to the primary drivers. While natural climate variation remains the

key driver of Mekong's hydrologic changes and variabilities over the last eight decades, the emergence of new dams has caused considerable changes to the river's hydrologic regime and flood characteristics in the past decade, which may have potentially profound implications on the ecosystem and livelihood of downstream areas.

Regarding the regional trend under natural climate variation, our results indicate that two main regions in the Mekong have generally changed over the last eight decades: the Lancang and the 3S basins. Overall, the trends suggest a substantial decline in river flow in the Lancang region, with values ranging from ~2.5-10% per decade, which can be seen in both annual volume and flow minima. The detected trend is prevalent in multi-decadal historical periods and, hence is likely to continue into the future, indicating that we might need to rethink dam operation and water management in this region. For example, the high reduction in annual minima in the Lancang region may lead to more water being held in the Lancang cascade dams, leading to dramatically reduced water levels in the downstream regions, especially during major droughts. Further, sustaining current hydropower production may become challenging in the future. In the 3S region, the tendency of increased (decreased) flow during the wet (dry) season indicates the potential for both increased flood risk and water scarcity. In this regard, existing dams may prove beneficial in mitigating floods and providing additional water in the dry season, if operated considering these changes as suggested in previous studies (Galelli et al., 2022; Pokhrel & Tiwari, 2022). However, such changes in flow patterns will have unintended consequences on downstream ecosystems and livelihoods (Arias, Cochrane, et al., 2014; H. Dang et al., 2022; T. D. Dang et al., 2016). Lastly, the decrease in annual minima in TSL's seasonally flooded areas suggests that the lake is becoming more stagnant, potentially contributing to less river flow to the downstream areas than in the past.

In terms of flood pulse, our results agree with previous studies (P. T. Adamson et al., 2009; Kummu & Sarkkula, 2008; Västilä et al., 2010a) that the average timing of the Mekong's flow regime has not drastically shifted during the past eight decades. The wet season typically occurs between June and November, while minimum and maximum flows occur in March and September, respectively. However, our results further reveal substantial fluctuations in the seasonal timing, sometimes exceeding 50 days per calendar year, and this is heavily dependent on the natural climate variation at each location. The results also show that while the onset of the wet season varies greatly over time, the end of the wet season has remained relatively stable. This means that the duration of the wet season each year can be predicted by how late the onset of the wet season was. Additionally, our results suggest that there is an abrupt shift of seasonal timing (~2-4 weeks) naturally occurs in areas downstream to station 11 compares to upstream locations, which can be directly attributed to the retention reservoir effect of the TSL. We also find that dam impacts on the seasonal timing of the Lower Mekong mainstream, specifically the MD, are relatively small, ranging between 2-5 days. However, the effect of accelerated dam operation has considerably delayed the wet season onset in the upper regions of the Mekong, sometimes by more than 30 days, especially in recent years. This can be particularly damaging to the environment as these impacts have been more pronounced in recent years when the Mekong was already witnessing a severe drought. This implies that dams are not mitigating extreme drought conditions in terms of seasonal timing but, in fact, are further worsening the delay of the wet season in upstream areas of the MRB.

On a similar note, our results suggest that the effect of mitigating extreme drought conditions through interannual water redistribution is relatively minor. While there are some effects of holding back water in one year and releasing in another, the difference due to dam operation is found to be only one-fourth of what natural climate variation can cause. However, dams are particularly effective in shifting water seasonally. Results (e.g., Fig. 6) illustrate that, since 2010s, there is a consistent shift by 10-20% of annual volume from the wet season to the dry season. While this impact is still more pronounced in the upstream, ~10% increase in most of the lower mainstream areas during the dry season is prevalent. Overall, this suggests that dams are causing the dry season to be wetter and wet season to be drier similar to previous findings (Piman et al., 2016; Räsänen et al., 2012), but less substantial than expected, especially in lower mainstream areas. While this effect can be positive in terms of agriculture as more water is available and is easily accessible for irrigation, it could cause a significant change and possibly irreversible adverse impact to the ecosystems. As also noted in previous studies (M. E. Arias, Cochrane, et al., 2014; M. E. Arias et al., 2013), water levels and inundated areas have increased in the dry season, causing many wetland areas to not have the dry period they need and could eventually destroy these important ecological systems. Our results further confirm that dams are negatively impacting the flow by reducing average flow and high flow in the floodplains of the TSL and MD. Further, due to the decreased high flow during the wet season, flood occurrence will be reduced in many downstream regions, especially outer areas of main water bodies, as also discussed in previous studies (H. Dang et al., 2022; Pokhrel, Shin, et al., 2018). However, due to the increase in low flow, flood occurrence in many areas of the TSL and Mekong near the main water bodies has been prolonged substantially by 10-15% (~36-54 days per year).

## 5. Conclusions

This study presents the first long-term (1940-2022) decadal trends and variabilities in river flow regimes over the entire MRB at a spatial resolution of ~5km. Historical changes in the seasonal timing and volume of the mainstream Mekong flow are examined and attributed to natural climate drivers and dam operation, with an emphasis on the temporal evolution of the Mekong's flood pulse. Then, the dam-induced impacts on the spatial-temporal changes in flow regime attributes of the TSL and MD are investigated by examining the decadal difference between simulations with and without dams for average flow, high flow, low flow, and flood occurrence over the last two decades. We draw the following key conclusions. First, natural climate variation caused substantial decadal trends (±5-10%) in river flow during 1940-2022 in portions of the MRB, especially the Lancang and the 3S regions. Second, dams are found to have intensified the high natural variability in seasonal timing of mainstream Mekong flow even though dam-induced impacts are still smaller compared to natural climate variation and typically more pronounced in areas directly downstream of dams. This can be observed in 2019 and 2020, during which dams exacerbated drought conditions by substantially delaying the MRB's wet season onset. We note that the wet seasons under natural conditions (simulation without dams) in these years were already alarmingly shorter than in average years (by 20-50 days). Third, upstream dam operation had minimal impact on annual flow but is largely altering the seasonality of MRB's flow regime attributes and flood dynamic in the TSL and MD by redistributing a substantial flow volume (~10-20%

annual volume) from wet season to the dry season; this is found to be substantial in the Mekong mainstream, especially in recent years. With reduced high flow in the Mekong mainstream, the decreased flood peak directly reduced flood occurrence (up to 5% or 18 days per year) in the surrounding floodplain areas. However, the increased low flow substantially prolonged the inundation of flooded areas in close proximity of the mainstream by ~36-54 days in some areas. As a result, dams have effectively reduced the typically extensive flooding in the TSL and MD, which could cause unprecedented adverse impacts to the ecological system and local communities. Our results might contain uncertainties caused by the use of basin-wide model, imperfect model parameterizations, uncertainties in input data including dam attributes with the use of a generic dam operation scheme. These uncertainties could have been further amplified by lacking considering of other human activities such as changes of land use, sand mining and water infrastructure development (e.g., dikes) that have been accelerating in recent times. These aspects could be addressed in future studies. Despite these limitations, this study presents the first results of the changes in natural hydrologic regimes in the MRB over the past eight decades, providing key insights on the role of recent increases in dam construction on changing annual/seasonal flow volumes, maximum flows, minimum flows, and flood occurrence in the TSL and MD areas. The findings are expected to be of use to rethink water resource management, especially in the face of climate change and planning of future dams, and open new avenues for research to address emerging dam-related issues in the MRB.

*Code and data availability*

Observed river flow and water level data can be obtained from the Mekong River Commission (http://www.mrcmekong.org/). ERA5 data are open access and are available at the Copernicus Climate Data Store (CDS; Hersbach et al., 2020; https://doi.org/10.24381/cds.bd0915c6, Hersbach et al., 2023). The Mekong dam list and CMFD code are available upon request. Simulated data used to produce the figures will be available on CUAHSI Hydroshare.

*Author contribution*. HD: Conceptualization, Data Curation, Methodology, Software, Formal analysis, Validation, Visualization, Writing – original draft preparation, Writing – review & editing. YP: Conceptualization, Funding acquisition, Resources, Writing – review & editing.

*Competing interests*. The contact author has declared that none of the authors has any competing interests.

*Special issue statement*.

*Acknowledgements*. This research was supported by the National Science Foundation (CAREER Award; Grant #: 1752729) and the Luce Foundation (LuceSEA – Funded Mekong Culture WELL Project). Simulations were conducted using HPCC provided by the Institute for Cyber-Enabled Research at Michigan State University.

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
