# Peer review of "Evolution of river regime in the Mekong River basin over eight decades and role of dams in recent hydrologic extremes"

_EGUsphere, 2023_

## Author Response (AR1)

**Editorial Comment:**

I have now received two expert reviewers' assessments and the corresponding proposed amendments by the authors. Both the reviewers agree on the potential for the manuscript to make a relevant contribution to HESS, yet they raised important points. The authors' planned changes are heading in the direction desired by the reviewers. Since not all suggestions provided by the reviewers seem implementable, it is important to clearly and comprehensively highlight the limitations of the proposed approach and, where possible, make an effort to add some supporting numerical analysis.

*Response: We are grateful to Editor Andrea Castelletti for handling our manuscript and providing helpful instructions to prepare the revised version of our manuscript. We have implemented the suggestions from the reviewers with additional analysis as well as providing notes that clearly highlight the current limitations in the proposed approach and possible improvements that can be made in future studies. We believe that the new version has benefited substantially from these revisions.*

**Referee Comment 1:**

The manuscript presents an interesting and thorough examination of natural variation and dam impacts on the Mekong's hydrological extremes over eight decades. The authors looked at a number of hydrological metrics such as flow trend, seasonal timings, flow volume, and extreme events and flooding patterns every decade. With the use of a hydrodynamic model, the authors simulated flows with and without dams so as to quantify the dam impacts and look at Mekong's hydrology under natural variability. Overall, the most significant contributions this study provided are (1) long-term trend of flow and (2) a full picture of natural flow and hydrological extremes, both drought and flood, compared to dam-altered flow and extremes in the whole Mekong Basin. In general, the manuscript in its current form is well written with insightful figures. That said, the manuscript can always use a bit of proof-reading and further synthesis.

*Response: We thank Dr. Mauricio Arias for taking the time to read the manuscript and providing constructive comments that incisively helped improve the quality and credibility of the manuscript. We have revised the manuscript and correlated our results with relevant findings from suggested sources and addressed other comments. A detailed, point-to-point response is provided in the following.*

While this study is certainly a step ahead of previous research in the Mekong looking at the effect of dams on river flows, the more comprehensive study results also means that the study ought to be controlled for two important factors which are directly linked to the conclusions of the study. First, while the study calibrated and validated the model in the conventional way, it is not clear to me that the authors are able to verify if the model captures the multidecadal variability and trends the study elaborates on. How can you ensure the reader that such term trends are real and not a model artifact? I suggest a stronger verification process, including how (or if) the study captures well-documented long-term variability and trends. See for instance the papers by Delgado (2010, 2012, the first already cited in the manuscript) and Rasanen (https://www.hydrol-earth-syst-sci.net/17/2069/2013/; 10.1016/j.jhydrol.2012.10.028).

*Response: This is a good point, thanks for raising it. To further prove the model's ability to capture the multidecadal variability and trends, we have provided additional validation of these characteristics against the available observed data at each selected station in Figures S7-9 of the supplementary and include additional analysis specifically on this in the model performance section of the revised manuscript in lines 231-234, page 10.*

Another aspect I do not think the study controlled against was water infrastructure in the floodplains and delta. As I hope the authors are aware, there has been widespread development of water control and irrigation infrastructure in both Cambodia and Vietnam, with effects on flooding well documented. Without taking a close look at what is happening with infrastructure within the floodplains, I would be extremely cautious about analysis and assertions made for those locations in Cambodia and Vietnam. The fact that the water level validation for these stations was not great may be a proof of this.

*Response: Thanks for raising this important point. Indeed, infrastructure development in the delta region is an important issue. While the massive system of channels transferring water across the Delta has been considered in our model as linkages between grid cells, its representation is partly limited due to its spatial resolution. Additionally, the current model version doesn't include the capability to account for dikes or other infrastructures. We are aware of those and have been working to account for these infrastructures in the model, which will be presented in our forthcoming articles. We believe that our findings would not be drastically different if we had accounted for these missing factors, especially considering that the analysis in Figure 6 shows the possible difference of flood occurrence in decadal terms instead of actual magnitude*

*while the remaining analyses are generally based on discharge. In the revised manuscript, we have added caution regarding any potential impact on our results caused by these factors in lines 370-372, page 16, and lines 516-518, page 22.*

In addition to these general comments, I have several punctual comments and suggestions throughout the manuscript:

Abstract:

1. Line 13: These are certainly not the first results of hydrological change in the Mekong. Consider deleting the word "first".

   *Response: We appreciate your comments, and we note that there have been previous studies providing critical insights into the hydrological change in the Mekong, some include long-term (~80 years) analysis such as Delgado et al. 2010. However, to the best of our knowledge, previous studies mainly analyze long-term observations at selected stations or basin scales in short time periods, and none have yet to provide an analysis at basin scale of this temporal coverage. To address this issue, we have added the term "basin-wide" to the abstract in line 13, page 2 to avoid causing any confusion.*

2. Lines 18-19: I suggest adding specific numbers associated with this claim that dams are intensifying variations in wet season flows.

   *Response: Thanks for the suggestion. We have added "…, the additional 65 dams commissioned since 2010 …" to the abstract in line 19, page 2 of the revised manuscript.*

Introduction:

3. Line 29: I would be cautious with the use of the word "stable", as many can interpret this as a "flat hydrograph", which is the complete opposite from what the subsequent references allude to.

   *Response: Thanks for the suggestions. We have changed the wording to "A consistent pattern of river regime…" in the revised manuscript, line 28, page 3.*

4. Lines 49-50: Here is another newer reference related to dam impacts in biodiverse rivers: Winemiller, K.O., et al, 2016. Balancing hydropower and biodiversity in the Amazon, Congo, and Mekong. Science 351, 128–129.

   *Response: Thanks for the suggestion. We have added this reference to the revised manuscript, line 43, page 3.*

5. Line 56: here you stay that the Mekong has "rather unpredictable seasonal variability". Do you refer to interannual variability? I argue that the Mekong river flow intraannual/seasonal variability is highly predictable (wet season in May-Nov, driest months Feb-Apr). The fact that you have hydrological models with very high fit to observations supports the predictability of this system.

   *Response: Thanks for the comment. While there is a general understanding that the Mekong river's wet season occurs during May-Nov, the actual onset and ending of it varies greatly (1-3 weeks) from year-to-year which is shown in Figure 4a. Additionally, the good fit from our model may*

*suggest that it could replicate historical conditions based on reanalyzed data (ERA5), however, there would still be considerable uncertainties in predicting future seasons/conditions that have yet to happened. We have modified this point to avoid confusion in the revised manuscript, lines 50-51, page 3.*

100

6.  Line 57: the statement here about seasonal timing sounds rather contradictory to the statement I talked about in the previous comment.

105     *Response: Thanks for the comment. We have addressed this in our response to comment #5 above.*

7.  Line 60: That the Mekong has a flood pulse dynamic has been documented for well over a decade, so consider updating this ref with one of the more historic ones:

Kummu, M., Sarkkula, J., 2008. Impact of the Mekong River flow alteration on the Tonle Sap flood pulse.
110     Ambio 37, 185–192.

Västilä, K., Kummu, M., Sangmanee, C., Chinvanno, S., 2010. Modelling climate change impacts on the flood pulse in the Lower Mekong floodplains. Journal of Water and Climate Change 01, 67–86.
https://doi.org/10.2166/wcc.2010.008

*Response: We have added the reference to the revised manuscript, line 53, page 3.*

115

8.  line 60: a reference is needed for the sentence "During the remainder of the year, river flow gradually reduces to less than 10% (sometimes 5%) of its flood peak."
    *Response: We have added Adamson et al., 2009 as the reference for this sentence in the revised manuscript, line 55, page 3.*

120

9.  Line 71: I'm confused about the mentioning of the Chi-Mun basin here, as this is a basin upstream of what is normally considered the Mekong floodplains. Besides, this region has had water infrastructure for much longer than the rest of the MRB, so if there is anywhere in the basin where the flood pulse has been affected for a long time (since the 1970s), its in the Chi Mun.

125     *Response: Thanks for the comment. We have decided to remove this mentioning in the revised manuscript, because we found that it doesn't add much value and could cause confusion.*

10. Line 82: Vastila et al (2010) looked primarily at inundation changes from climate change. A more relevant ref to the ecological changes would be Arias et al (2014b):

130  Arias, M.E., Cochrane, T.A., Kummu, M., Lauri, H., Koponen, J., Holtgrieve, G.W., Piman, T., 2014.
Impacts of hydropower and climate change on drivers of ecological productivity of Southeast Asia's most important wetland. Ecological Modelling 272, 252–263.

*Response: We have replaced the reference in the revised manuscript, lines 79-80, page 4.*

135  11. Line 83: I suggest stating "hydropower development" instead of "booming hydropower dams"
    *Response: This is a good suggestion. We have changed the term to "rapid hydropower development" in lines 80-81, page 4, as we want to emphasize the rate of change that is happening.*

Data and methodology:

12. Data: Given the poor relationship we know exists between water level observations and derived discharged, I would be very cautious with analysis based from results for stations 10-13, for both levels and discharge. In addition, stations 12 and 13 are strongly influenced by water infrastructure in the Delta, which is not considered in this study.

*Response: Thanks for the comment. We agree that there could be many uncertainties arising from the poor relationship between observed water level and derived discharge. However, our aim is to highlight the long-term changes of water balance and separate dam impacts from natural drivers as well as other human factors. Indeed, analysis based on water level would have more uncertainties as the river morphology can change and has been changing drastically in recent years due to dam trapping and sand mining. Thus, our analyses focus mainly on long-term average discharge would eliminate these uncertainties. In the revised manuscript, we have added notes about this point along with the discussion on missing infrastructure in the Delta in lines 370-372, page 16.*

13. Lines 156: Though I am aware that this model has been documented multiple times before, it might be informative to mention the basic hydrodynamic assumptions the model use (e.g., kinematic wave?)

*Response: This information has been provided in lines 159-160 of the original manuscript (154 in the revised version) and we have added reference to the model's development paper (Yamazaki et al., 2013) in the revised version, line 155, page 7, in case readers seek further information.*

14. lines 165-166: is it CFD or CMFD? I think there is a typo here. I would discourage you from using the acronym CFD, which is a very common acronym used in water science and engineering for Computational Fluid Dynamics.

*Response: We have changed all CFD to CMFD in the revised manuscript.*

15. Line 177: Just to clarify: Your model switches on dams only after their corresponding year, is that correct? How did you consider reservoir infilling?

*Response: Thanks for this note. Considering that the actual date of reservoir operation/filling is generally not available or collected widely, we believe it is a good assumption that the dams start filling even before the official commissioned dates. Thus, our model switches on dams at the beginning of the reported commission year and starts operating from that point instead of the end of that year. We have added clarification in the revised manuscript, line 179, page 8.*

16. Lines 198/Figs S2-3: Though I see that fig S2-5 present time series of ERA5 vs biascorrected, having 12 frames per figures, each with 20+ years of time series, makes it really difficult to truly see the difference between the simulations. I recommend a simpler graph illustrating the biascorrection process for a single station and shorter timeframe.

*Response: We have added Fig.S2. in the supplementary, showing comparison of simulated results before and after bias correction, with and without dam for a shorter timeframe in the revised version, and added discussion on this in lines 199-200, page 8.*

17. I notice that statistical indicators for model performance were not mentioned in the Methodology section. They instead first appeared in the Results section. I suggest a mention of those indicators in Methodology.

*Response: We have added additional sentences mentioning the statistical indicators in the data processing section of the revised manuscript, lines 204-206, page 9.*

Results:

18. Model performance: What about testing for interannual variability and trends? Because the model predicts well seasonal variability, it does not mean that it also reproduces well how those patterns varied from year to year. Later on in the results you use the model results to make assertions on interannual variability and trends, but perhaps early on you have to demonstrate that your model (and input reanalyzed data) is in agreement with those trends that have been found from observations alone.

*Response: Thanks for raising this point. To further prove the model's ability to capture the multidecadal variability and trends, we have provided additional validation figures of long-term trends for average, minimum, and maximum river flow in the supplementary comparing these characteristics between observed and simulated data at each selected station during the period when the observed data is available and additional notes on these in the revised version, lines 231-234, page 9.*

19. line 221: Please correct the typo, Stung not Strung.

*Response: Thanks for a very detailed reading. We have corrected this in the revised manuscript, line 224, page 9.*

20. Lines 221-223: I am a bit concerned that the water level simulations provide worse results than discharge. This probably has to do with the river bathymetry representation used in the model. What did you use by the way? Perhaps I missed that from the methods. I am sure that the authors realize that discharge measurements are seldom made, and that the vast majority of the discharge observations in the Mekong are derived from water level observations via regression equations. Thus, if there are inconsistencies in the water level measurements (which I do not think is the explanation here) there are certainly inconsistencies with the discharge data. The Mekong floodplain becomes really flat south of Kratie, so we should be very skeptical of any discharge data calculated (not observed) for the lower four stations.

*Response: The river-floodplain topography parameters used in our simulation are derived from the MERIT Hydro dataset (Yamazaki et al., 2017) and we have noted this in section 2.2, line 153. Indeed, the discrepancy between observed and simulated water levels was due to river bathymetry representation which could be caused by the following reasons: (1) river-floodplain topography parameters are fixed over the entire simulation period, (2) uncertainty from the original satellite product that was used to derive the MERIT Hydro dataset. However, the model considers both flow in river and the floodplain, thus, while the representation of the river bathymetry might not be highly accurate during the validation period, we believe the water balance of the area is reasonably accurate for further analyses. We have noted these issues in the revised version, lines 225-231, page 9.*

21. Lines 254-255: Here you state that the Lancang has experienced a decreasing trend, but do not really explain why. This is contrary to what has been reported for rivers starting in the Himalayas, where icemelt has increased flow in the upper parts of the Mekong and other rivers in the region: Li, D. et al. High Mountain Asia hydropower systems threatened by climate-driven landscape instability. Nat. Geosci. 15, 520–530 (2022).

*Response: We appreciate the reviewer's comment, and we did examine the paper carefully, however, we are still struggling to find a substantial connection between this study and the current manuscript, thus, we appreciate it if the reviewer could provide more detailed information or suggestions on this.*

22. lines 273-274: there's a repetition of the word 'increase'. Please fix.
*Response: Thanks for your thorough reading. We have removed the duplicated word in the revised version.*

23. Line 288: Regarding the comment about natural retention in the Tonle Sap, Stations 12 and 13 are deep into the Delta in Vietnam, quite a ways from the Tonle Sap, but in the middle of one of the largest rice production regions in the world. **Thus, you need to show some strong evidence to support that the Tonle Sap retention has effects there greater and the massive and active water infrastructure network Vietnam has built in the Delta.** This might be related with the calibration results for these stations being below optimal.
*Response: Thank you for raising this point. We have provided the additional Figure 5 on water balance analysis and seasonal timing comparisons between station 11 (upstream of Phnom Penh), Prek Kdam (Tonle Sap River), and other locations on the Mekong mainstream that are directly downstream from Phnom Penh in the Mekong Delta to provide additional evidence with major notes on the impact of Tonle Sap as well as the Mekong Delta channel effects in the revised version, lines 337-372, pages 14-15.*

24. line 317: "dams are generally delaying the wet season onset…". The author should expand that part by discussing the wet season duration because the example inside the brackets is about duration, not onset.
*Response: We have modified the example to wet season onset in lines 325-326, page 14.*

25. line 318: based on Figure 4e, I think longer duration is not just in the early 2000s but in the 2000s as seen in 2001, 2005, 2007, and 2009.
*Response: Indeed, the longer duration happened in some of the upstream locations during the mentioned years. We have added a sentence mentioning this more specifically in the revised version, lines 326-327, page 14.*

26. Figure 5: why not include a figure of 'Natural anomaly of dry season volume'? This should be consistent with how Figure 4 was depicted.
*Response: Thanks for raising this point. Indeed, the analysis of dry season volume anomaly would provide a meaningful insight if the seasonal timing were fixed on specific dates each year. However, the current version of Figure 5 dry season volume is based on the detected timing in Figure 4, thus, the total number of days in each dry season would be slightly different. As a result, the total volume*

*in each dry season can be considerably different due to timing changes (as shown below) and provide little additional meaning to the current analysis as shown in panel (c) below. Thus, we have decided to keep Figure 5 (Figure 6 in the revised version) as in the original manuscript.*

[Figure]

275

27. Lines 367-370: This sentence here seems like a long way of describing what I think is an obvious logic (a decrease in peak flow decreases flow in the floodplains). Consider deleting or synthesizing.

*Response: While it is generally understood that a decrease in peak flow lowers the flow in the*
280 *floodplains, we wanted to highlight that this decrease is due to dam impact. Furthermore, this sentence was meant to contrast the findings in the following sentences which is that an increase of flood occurrence in the inner areas is also caused by dam operation. Thus, we have incorporated that sentence and the next one in lines 414-415, page 18 to make a direct connection from the changes of flow to flood occurrence.*

285

Discussion:

28. Lines 403: with regards to changes in the Lancang and 3S basins, it would probably be good to compare and contrast your results with those studies that have looked at dam development specifically for those regions. For example:

Räsänen, T.A., Koponen, J., Lauri, H., Kummu, M., 2012. Downstream Hydrological Impacts of Hydropower Development in the Upper Mekong Basin. Water Resources Management 26, 3495–3513. https://doi.org/10.1007/s11269-012-0087-0

Piman, T., Cochrane, T.A., Arias, M.E., 2016. Effect of Proposed Large Dams on Water Flows and Hydropower Production in the Sekong, Sesan and Srepok Rivers of the Mekong Basin. River Research and Applications 32, 2095–2108. https://doi.org/10.1002/rra.3045

*Response: Thank you for your suggestion. We have added notes on consensus findings and cited the suggested papers in the revised version, lines 483-484, page 21.*

29. With regards to the flood pulse, there is a newer paper that is probably worth reviewing:

Morovati, K., Tian, F., Kummu, M., Shi, L., Tudaji, M., Nakhaei, P., Alberto Olivares, M., 2023. Contributions from climate variation and human activities to flow regime change of Tonle Sap Lake from 2001 to 2020. Journal of Hydrology 616, 128800. https://doi.org/10.1016/j.jhydrol.2022.128800

*Response: We have reviewed the suggested paper and added additional linkage on relevant findings in line 346, page 14.*

Conclusions:

30. Lines 452-3: As mentioned earlier, this conclusion can only be drawn if you verify that your model effectively can replicate long-term observations.
    *Response: We have provided additional validation on trends as noted above.*

31. Line 459: Remove "or higher" to avoid confusion.
    *Response: We have removed it in the revised version.*

32. Line 467: the use of static land cover is also a limitation, despite arguably not having substantial impacts on flow alterations compared to dams.
    *Response: We have added this in the revised version, line 517, page 22.*

**Referee Comment 2:**

The authors investigated the long-term trend and variability of hydrological extremes in the Mekong River basin under natural conditions as well as under the influence of growing dam development. The investigation is based on 83 years of hydrological data, simulated by a hydrodynamic model with and without representing the dams. Although quite a few previous studies investigated the Mekong's hydrological regime, the use of relatively long hydrological data in this study has provided useful insights for the river's water management that has multi-national and multi-sectoral importance. The paper is well-written with nice visualizations. I am overall in favor of the publication of this paper. That said, I agree with the comments by Reviewer-1, and add the following comments that could help improve the clarity and readability of the paper.

*Response: We thank the referee for spending time reviewing our manuscript and providing constructive comments that helped improve the quality and clarity of the manuscript. We have revised the manuscript and provide additional information based on your thoughtful comments. A detailed, point-to-point response is provided in the following.*

**Major comments:**

1. **Introduction:** The first two paragraphs in the Introduction describe the general effects of human intervention on the globally important river basins. This is a bit too long. The authors could shorten this discussion and focus on the problem statement and research gap around the Mekong. In particular, given that Mekong's hydrologic regime has been studied by several other studies, I was not clear about the specific contribution of this study until reading the last couple of paragraphs. Therefore, revising and reshuffling the ideas in the Introduction could improve the readability of the paper.

   *Response: Thanks for your recommendation. Indeed, we recognized that the general information at global scale in the first two paragraphs could be too long and it takes considerable reading effort to have a clear understanding of our studies contribution. Thus, we have shortened the Introduction section by incorporating the first two paragraphs.*

2. **Methods (line 160):** Why is the water released from multi-purpose dams set to optimize power generation? I suppose some large-storage dams in the Mekong prioritize water availability for irrigation rather than power generation.

   *Response: Thank you for raising this, it is a good point. While we agree that multi-purpose dams can be prioritized for water availability for irrigation, water supply, or flood control instead of hydropower generation, this priority can be changed periodically based on local demand and government planning, which information is generally not available. In an ideal situation, these dams' operation should be controlled to meet different priorities in different season/flow situations, which we are working on to account for in our forthcoming studies. It should be noted that all 3 multi-purpose dams considered in our study are reported to have hydropower generation capacity. Additionally, while dams optimized for power generation might not meet the entire irrigation demands downstream, this demand should be partially satisfied from continuous dam release. Thus, we believe that our findings would not be drastically different if we had set the multi-purpose dams*

*to operate as irrigation dams. In the revised manuscript, we have added additional clarification on this in lines 158-159, page 7.*

**Minor comments:**

**3.** Line 10: maybe 'in the last few decades' instead of 'in the last decade'.

*Response: Here, we specifically aim to highlight the 2010-2020 period considering that the basin-wide storage capacity has increased by almost 3 times compared to before that as shown in Figure 1.*

**4.** Lind 15-16: maybe replace 'has undergone' with 'shows'.

*Response: Thank you; we have edited this in the revised manuscript, line 16, page 2.*

**5.** Line 16 (and similar use thereafter): The use of 'decadal trends and variabilities' is not that clear. Are you referring to the changes/variability under large-scale climate drivers like ENSO?

*Response: Yes, we were referring to the changes/variations under natural climate drivers such as El Nino, La Nina, etc.*

**6.** Line 21: Are 2019 and 2020 examples of normal years or years of shorter wet seasons?

*Response: Thank you for your comment. Those years are meant as examples of the years with shorter wet seasons. We have edited this in the revised manuscript, line 20, page 2.*

**7.** Figure on page 2: What is on the y-axis in the graphs? Also, you used the same color palette for different data on the maps and graphs which could be confusing. Maybe add a caption too.

*Response: Thank you for your suggestions. The figure on page 2 was meant as a graphical abstract, which is specifically instructed by Hess author guidelines as unnumbered and placed directly after the text abstract. Additionally, the figure is combining highlights from selected results in our paper,*

*thus we have decided to keep the color scheme similar to the main figures to avoid confusions. The numbers on the y-axis of the top middle and right panels are the numbering order of the selected stations, which, we have added axis title in the revised version.*

**8.** Line 36-37: Dams in general could benefit flood control but a majority of the Mekong's dams are for hydropower. Maybe it's worth mentioning that.

*Response: Thank you; we have added this information in the revised version, line 71-73, page 4.*

**9.** Line 44: 'desiccation' seems a less commonly used word. Maybe change it.

*Response: Considering that this is frequently used in scientific literatures related to the Aral Sea, we have decided to keep the term as is in the revised version.*

390  **10.** Line 50: The year for Zarfl et al. should be 2015.

*Response: Thank you for your thorough review. We have fixed this in the revised version, line 43, page 3.*

**11.** Line 67-68: Do you indicate the 'ecosystem services' by reliance on 'local communities' on flood pulse? Also, 'have developed to' probably does not fit to this sentence.

395  *Response: Here we refer to the long-term natural development of both ecosystem services (i.e., fish migration, wetlands growth, etc.) and local communities' culture and livelihood (fishery, rice cultivation) based on the timing of the Mekong's wet season, which onset is signified by the flood pulse. We have edited this to provide more clarity in the revised version, line 62, page 4.*

**12.** Line 69-75: I suppose fish production is another important ecosystem service that could be
400  mentioned here. Mekong is one of the key hotspots in the Global South where strong tension exists between freshwater resources and dam development for energy-economic growth. See the recent paper by Chowdhury et al. (2024): Hydropower expansion in eco-sensitive river basins under global energy-economic change. Nature Sustainability. DOI: https://doi.org/10.1038/s41893-023-01260-z

405  *Response: Thank you for your suggestions. We have added fish production and reference the suggested paper in lines 72-73, page 4.*

**13.** Line 103: maybe replace 'flood researches' with 'flood-related studies'.

*Response: We have replaced this in the revised version, line 100, page 5.*

**14.** Line 154: What is the spatial resolution of the unit catchment? Could each unit catchment represent
410  one or more reservoirs? Could some large reservoirs span over multiple unit catchments? A visual presentation (at least in SI) of the unit catchment could be helpful.

*Response: The spatial resolution of each unit catchment in our simulation is 5km, additional information on the unit catchment representation can be found in Yamazaki et al., 2017 as noted in line 153, page 7. Currently, the applied operation scheme only allows the representation of one
415  reservoir in each cell. There was only one case that two dams (release gate) are in one same cell, in which, we've considered to keep the one with larger storage capacity for simulation. Indeed, some of the reservoirs span over multiple unit catchments in our simulation. Lastly, thank you for your suggestion, we have added Fig.S15 in the supplementary showcasing the reservoirs extent in the revised version.*

420  **15.** Line 165: I agree with R1 about not using the "CFD" acronym here.

*Response: Yes, we agree, and we have changed all "CFD" in the manuscript to " CMFD".*

---

## Author Response (AR2)

**Editorial Comment:**

The authors have done a good job revising the manuscript. Only a few minor pending points remain unsolved and I will check their amendment with no further need of review.

*Response: We really appreciate Editor Andrea Castelletti for handling our manuscript. We have revised the manuscript and prepared the response to reviewer's comments in the following.*

**Anonymous Referee #1 (Report #2) comments:**

I reviewed this manuscript during the first round and I see that the authors have addressed all comments brought up by both reviewers. Well done. No major technical issue remain, just a few minor suggestions:

1. L 323: Remove "the" in "construction of dams in the recent decades"

*Response: We have removed "the" in line 323, page 14.*

2. L 339: "Retention reservoir" sounds a bit redundant, plus we know that the TSL is not a reservoir in the way is generally discussed in this paper. I suggest changing here and through the paper to "natural retention effect".

*Response: We agree with the referee and have removed "reservoir" in line 339, page 14.*

3. L 341: Fix the text in "between 11 and 12, 16". Perhaps better "between 11 and 12/16"?

*Response: Thank you for the suggestion. We have changed the text to "between 11 and 12 (or 16)" for more clarity in line 341, page 14.*

4. L 469: Same as in L 323.

*Response: We have modified the text from "retention reservoir effect" to "natural retention effect" in lines 469-470, page 20.*

5. The Ziv et al 2012 reference is duplicated.

*Response: Thank you for your thorough reading. We have removed the duplicated reference.*